# Cooperation by necessity: condition- and density-dependent reproductive tactics of female house mice

Manuela Ferrari [1,3 ✉], Anna K. Lindholm [2], Arpat Ozgul [2], Madan K. Oli[1] & Barbara König [2]

Optimal reproductive strategies evolve from the interplay between an individual's intrinsic state and extrinsic environment, both factors that are rarely fixed over its lifetime. Conditional breeding tactics might be one evolutionary trajectory allowing individuals to maximize fitness. We apply multi-state capture-mark-recapture analysis to a detailed 8-year data set of free-ranging house mice in a growing population to discern causes and fitness consequences of two alternative reproductive tactics in females, communal and solitary breeding. This allows us to integrate natural variation in life-history traits when analysing the expression of two alternative reproductive tactics in females. We find that communal breeding reduces average population fitness, but nevertheless increases over our 8-year study period. The tactic proves to be expressed conditionally dependent on both population density and female body mass – allowing females to breed under subpar conditions, *i.e.* at high density or when of low body mass. Our results contradict previous laboratory studies and emphasize the importance of studying cooperation under natural conditions, including natural variation in state-dependent survival and breeding probabilities.

---

[1] Department of Wildlife Ecology and Conservation, University of Florida, Gainesville, Florida, USA. [2] Department of Evolutionary Biology and Environmental Studies, University of Zurich, Zurich, Switzerland. [3]Present address: Mammalian Behaviour & Evolution Group, Institute of Infection, Veterinary and Ecological Sciences, University of Liverpool, Neston, UK. ✉email: manuela.ferrari@liverpool.ac.uk

Alternative reproductive phenotypes are ubiquitous among animals. In many invertebrates and vertebrates, individuals use alternative ways to optimize their reproductive success. Such alternative reproductive strategies are defined as discrete morphological, physiological or behavioral differences among individuals from the same sex and population[1,2]. Variability in reproductive phenotypes can arise and be stabilised by alternative tactics resulting in equal fitness or through condition-dependent tactics with different fitness optima for different conditions[1].

Studies that aim to understand such phenotypic variation usually measure individual annual or lifetime reproductive success[3–5]. In field studies on social insects, birds and mammals this approach has been used to analyse the conditions under which individuals switch between either breeding tactics or helping others to raise their young[4,6–9]. Lifetime reproductive success is an important component of fitness, but does not incorporate central aspects of the life cycle, such as the timing of the onset of breeding or age-dependent differences in survival and breeding probabilities. Ideally, we need a holistic demographic approach, which allows for a more robust estimation of the individual fitness effect at the population level. Wild populations with natural variation in life-history traits would be especially valuable in understanding the interplay of the different factors and how they affect fitness. Datasets from free-living species, however, are rarely detailed enough to permit such analyses.

Here we use long-term data (8-years) from a population of free-ranging house mice (*Mus musculus domesticus*) living in a barn in Switzerland, where individuals are followed from birth until death or until they disperse out of the building, with detailed information on behavior, survival and reproductive success. We aimed to quantify population-level effects of two plastic reproductive phenotypes in females, raising a litter either solitarily or communally. In the latter case, a female pools her litter with that of one or several other females. All mothers indiscriminately nurse own and other offspring when communally raising litters[10–12].

Communal breeding has been described for house mice under various conditions[13–15]. Females can switch between communal and solitary breeding in successive breeding attempts, suggesting that breeding tactics are not genetically fixed traits, but are phenotypically plastic tactics, allowing females to choose one or both tactics during their lifetime[5]. Various benefits and costs are associated with communal breeding, and in line with joint breeding in insects, birds and other mammals, it is considered to be a prominent example of cooperation[16–20]. In house mice, it provides mutualistic benefits by improving lifetime reproductive success of females communally raising their litters when kept under laboratory conditions[21], and increases pup survival due to better nest defence against both male and female intruders from neighbouring territories[14,22]. On the other hand, communal breeding can be costly due to within-nest infanticide and unequal benefits for the females if their litter sizes or amount of investment differ[12,23,24]. House mice associate spatially with related individuals[25,26], and it is conceivable that they are more likely to breed communally with related females. Indeed, breeding communally with close relatives can confer indirect fitness benefits or reduce some of the costs of potential exploitation[21,25,27]. While communal breeding has been shown to occur among both unrelated and related individuals, there is evidence that females have a higher probability to communally breed in groups with higher average relatedness[28]. Furthermore, female body mass - an intrinsic factor - affects breeding behavior in house mice. Females produce more milk with increasing body mass[12,29] and are likely to be dominant over smaller females in aggressive encounters[30].

Taken together, the observed plasticity in a female's breeding tactic might be influenced by her intrinsic state (her reproductive competitiveness) as well as by extrinsic environmental conditions (population density). Communal breeding might therefore be beneficial only under certain conditions, which, however, are yet to be ascertained in free-living populations.

We expected females to have an overall higher probability to reproduce and to raise more of their litters solitarily at low population density, because of low reproductive competition and limited opportunities for communal breeding[31]. At high density, increased competition and more females breeding at the same time[32,33] is expected to result in an overall reduction in females' breeding probability and at the same time a larger proportion of litters raised communally. Under such conditions, females of high body mass (*i.e.* in good condition) might be more competitive and have higher reproductive success.

In our study population, food, water, and nesting material were provided *adlibitum*, and the number of nest boxes was held constant ($n = 40$). Females preferentially raise litters in nest boxes and can breed all year round, with nevertheless pronounced seasonal differences[34]. Over the 8-year study period, population size increased from around 75 adults in the summer of 2007 to 265 in the summer of 2014, which allows us to analyse the effects of population density on breeding behavior across a wide range of density conditions. We used a multi-state capture-mark-recapture (MSCMR) model[35–37] and estimated state-specific female survival probabilities, as well as the transitions among life-history states. We further calculated season- and breeding tactic-specific litter sizes. Using these parameters, we constructed and analyzed season-, density- and breeding tactic-specific matrix population models[38], which allowed us to estimate population growth rate (a measure of average population fitness), and sensitivity of population growth rate to changes in life-history parameters. We thus assessed the fitness and population-dynamic consequences of alternative breeding tactics using relevant life-history traits, including state-specific survival and breeding probabilities, as well as litter size.

## Results and Discussion

Multi-state capture-mark-recapture (MSCMR) analyses revealed that over the 8-year study period female survival probability varied strongly depending on age, reproductive status, and season (Tables S1 and S2). Juveniles experienced lower survival than adults in all breeding states (Table 1). The probability of breeding (conditional on survival) in the next month was lowest for the prebreeders (females older than one month that never bred before) and slightly higher for females with prior breeding experience (females that were currently breeding or had bred previously but not in the current month (nonbreeder), Table 1). During October to February (off-breeding season), breeding probabilities were close to zero for all breeding states (Table 1). The state, season and density-dependent capture probabilities are given in SI files, Fig. S2.

We used the estimated life-history measures to parameterise seasonal matrix population models (Fig. 1). During the breeding season, monthly asymptotic growth rate $\lambda$ was 1.04 [95% CI, 1.03–1.05], indicating a growing population; the population was relatively stable during the off-breeding season ($\lambda = 0.99$ [0.98–1.00], both $\lambda$ values evaluated at stable stage structures). The annual asymptotic population growth rate was 1.21 [1.12–1.34], which corresponded closely to the observed growth rate of our study population. Sensitivity analyses revealed that $\lambda$ was most sensitive to changes in survival and breeding probabilities of the prebreeders and nonbreeders during the breeding season, while during the off-breeding season survival of prebreeders had the largest effect on $\lambda$. Sensitivities of $\lambda$ to changes in the probability of breeding communally were negative for all

**Table 1 State-specific monthly survival and breeding probabilities.**

| state | survival probability | breeding probability | communal breeding probability |
|---|---|---|---|
| $i$ | $\sigma_i$ | $b_i$ | $c_i$ |
| A) breeding season | | | |
| J | 0.40 [0.38-0.42] | 0 | 0 |
| P | 0.87 [0.86-0.89] | 0.16 [0.14-0.18] | 0.73 [0.65-0.80] |
| S | 0.93 [0.92-0.94] | 0.35 [0.27-0.46] | 0.46 [0.35-0.60] |
| C | 0.93 [0.92-0.94] | 0.33 [0.28-0.40] | 0.65 [0.56-0.76] |
| N | 0.93 [0.92-0.94] | 0.29 [0.25-0.34] | 0.72 [0.63-0.82] |
| B) off-breeding season | | | |
| J | 0.71 [0.64-0.76] | 0 | 0 |
| P | 0.97 [0.96-0.98] | 0.04 [0.03-0.05] | 0.80 [0.64-0.98] |
| S | 0.90 [0.88-0.92] | 0 | 0 |
| C | 0.90 [0.88-0.92] | 0 | 0 |
| N | 0.90 [0.88-0.92] | 0.05 [0.03-0.07] | 0.53 [0.34-0.81] |

State-specific monthly survival and breeding probabilities [95% CI] during the breeding season (Mar-Sep) and off-breeding season (Oct-Feb) estimated with the most parsimonious multi-state capture-mark-recapture model (MSCMR, see Table S1).
Survival probability is the probability that females that are currently in state i survive until next month; breeding probability is the probability that females that are currently in state i will breed next month (i.e., produce a litter) either solitarily or communally, conditional on survival; and communal breeding probability is the probability that females that are currently in state i will breed communally, conditional on a female being alive and breeding next month. For a detailed description of life-history states, possible transitions among states and an explanation of the abbreviations, see Fig. 1, $n = 4003$ females.

adult states (*i.e.* an increase in the proportion of communal litters would result in a decrease of the population growth rate $\lambda$), suggesting that in our population communal breeding results in lower fitness for females than solitary breeding. Nevertheless, we found that the proportion of litters raised communally almost doubled during the 8-year study period (Fig. S1).

**Density-dependence of life-history parameters**. We found strong evidence that population density influenced both survival and breeding probabilities; a model that included population density as covariate for both survival and transition probabilities was better supported than other competing models (Table S1, SI files). Population density had little or no effect on the survival of adult females in the breeding season (Fig. 2a), but improved survival of juveniles and prebreeders (Fig. 2a). The positive effects of population density on survival of juveniles and prebreeders coincide with a study conducted on dispersal in the same population[39], but contradict findings of earlier work that reported a negative influence of increasing population density on survival in several mammalian species[40,41], including house mice[33]. Capture-mark-recapture analyses do not distinguish between dispersal and death, so an alternative explanation for our results could be that juvenile dispersal decreased with population density. Although dispersal is generally thought to increase as population density increases[42], an opposite effect might be expected if the density is high both at the natal and potential dispersal sites[43,44]. Both young males and females disperse in house mice[45]; changes in dispersal rates would therefore mainly affect juveniles and prebreeders. Increased predation outside the barn population could potentially select against dispersal and consequently lead to a reduction in the number of disappearing juveniles and prebreeders over time. During the off-breeding season, juvenile survival was still increasing at higher densities, but the effect could no longer be observed for prebreeding females. On the other hand, the survival of adult females slightly decreased with increasing density (see Fig. S3, SI files).

In agreement with our initial expectation, female breeding probability decreased with increasing population density; adult females (except nonbreeders, i.e., females that bred previously but have no current litter) were less likely to breed as the population density increased (Fig. 2c). These results are consistent with observations that breeding probability and offspring survival are negatively affected by population density in house mice[32,33] and other small mammals[41,46].

Litter sizes decreased with increasing population density (slope parameter, $\beta$ [95% CI] = $-0.35$ [$-0.48$– $-0.23$]), with communal and solitary litters being similarly affected by variation in population density. Communal litters were consistently smaller than solitary litters (mean difference, $\beta = -0.22$ [$-0.34$– $-0.10$]). This is likely a consequence of increased within-nest infanticide in communally raised litters under high-density conditions, which regularly occurs in free-ranging and laboratory populations of house mice[5,21,23,24]. There was no evidence that communal breeding evolved as a mechanism to reduce infanticide by individuals from other groups, which could have potentially compensated for increased within-nest infanticide in communally raised litters. Were this the case, the difference in average litter size between solitary and communal litters should have decreased or disappeared altogether at high population density.

To examine the density dependence of a life-history parameter's effect on the population growth rate, we estimated state-specific survival and reproductive parameters at low (81 adult mice) and high (224 adult mice) population densities. Those densities were chosen to be close to the first and third quartile of all observed densities. We then constructed and analyzed seasonal matrix population models as described previously. At low population density, the house mouse population increased during both the breeding ($\lambda = 1.04$ [1.03–1.06]) and off-breeding seasons ($\lambda = 1.03$ [1.01–1.05]), as well as annually ($\lambda = 1.62$ [1.39–1.88]; always evaluated at stable age structures). $\lambda$ was most sensitive to changes in the survival and breeding probabilities of prebreeders. The sensitivity of $\lambda$ to the monthly probability of raising a litter communally (conditional on breeding) did not differ from zero (sensitivity [95% CI], prebreeder = $-0.001$ [$-0.004$–0.002], solitary breeder = $-0.001$ [$-0.004$–0.002], communal breeder = $-0.001$ [$-0.005$–0.003], nonbreeder = $-0.001$ [$-0.005$–0.003], Fig. 3, Table S4, SI files). At high population density, seasonal differences in population growth were more pronounced (breeding season: $\lambda = 1.04$ [1.03–1.05]; off-breeding season: $\lambda = 0.98$ [0.97–0.99]), resulting in the annual $\lambda$ to be ~25% lower ($\lambda = 1.20$ [1.04–1.33]) than at a low population density. In contrast to the low density population, the sensitivity of $\lambda$ to the monthly probability of raising a litter communally (conditional on breeding) was less than zero (sensitivity [95% CI], prebreeder = $-0.004$ [$-0.006$– $-0.001$], solitary breeder = $-0.001$ [$-0.001$– $-0.000$], communal breeder = $-0.002$ [$-0.003$– $-0.001$], nonbreeder = $-0.006$ [$-0.010$– $-0.002$], Fig. 3, Table S4, SI files)

**Density-dependence of breeding tactics**. We further expected an increase in communal breeding at high densities either because of fewer opportunities for solitary breeding (fewer nesting sites without a litter already present) or more potential partners available (more opportunities to join another litter). Consistent with this expectation, the probability to raise a litter communally increased as population density increased for all breeding states. Solitarily breeding females had the highest probability of raising another solitary litter the following month, conditional on survival. However, this effect diminished as the population density increased (see Fig. 2d). At high densities, the probability to raise the next litter communally or solitarily were very similar for

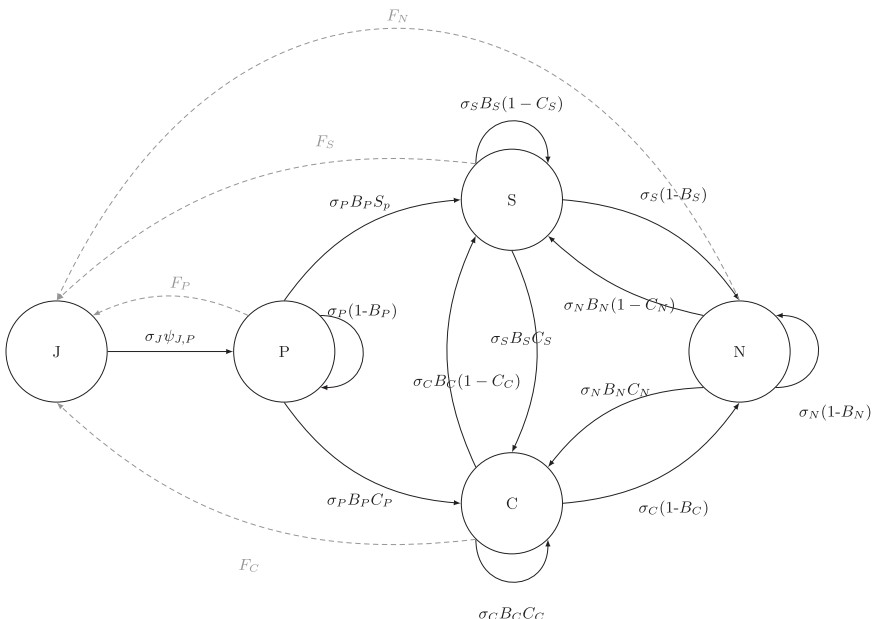

$$A = \begin{bmatrix} 0 & F_P & F_S & F_C & F_N \\ \sigma_J \psi_{J,P} & \sigma_P(1-B_P) & 0 & 0 & 0 \\ 0 & \sigma_P B_P(1-C_P) & \sigma_S B_P(1-C_S) & \sigma_C B_C(1-C_C)\psi_{4,3} & \sigma_N B_N(1-C_N) \\ 0 & \sigma_P B_S C_P & \sigma_S B_P C_P & \sigma_C B_C C_C & \sigma_N B_N C_N \\ 0 & 0 & \sigma_S(1-B_S) & \sigma_C(1-B_C) & \sigma_N(1-B_N) \end{bmatrix}$$

**Fig. 1 Monthly life-cycle graph for female house mice.** Each circle represents a life-history state, and each arc or arrow represents a transition from one state to another. Life-history states are: J = juvenile (first month of age), P = prebreeder (second month or older, not yet breeding), S = solitary breeder (female has a solitary litter), C = communal breeder (female has a communal litter), N = nonbreeder (female has no current litter, but bred previously). A post-breeding census was assumed. Survival probabilities are denoted by $\sigma_i$, and transition probabilities by $\psi_i$ (conditional on survival), with $B_i$ being the probability to breed (transition towards the breeding states: $B_i = \psi_{i,S} + \psi_{i,C}$) and $C_i$ the probability to breed communally, conditional on breeding ($C_i = \frac{\psi_{i,C}}{B_i}$). The fertilities were given as $F_i$ and were calculated the following way, with $f_i$ being the average number of female pups produced per litter for a given state: $F_P = \sigma_P B_P(1-C_P)f_S + \sigma_P B_P C_P f_C$; $F_S = \sigma_S B_S(1-C_S)f_S + \sigma_S B_S C_S f_C$; $F_C = \sigma_C B_C(1-C_C)f_S + \sigma_C B_C C_C f_C$; $F_N = \sigma_N B_N(1-C_N)f_S + \sigma_N B_N C_N f_C$. Note that females that are currently in a nonbreeding state can reproduce (and thus, have a non-zero fertility rate), because some of them will survive and reproduce before the end of the calendar month (*i.e.* the end of the session.).

females in all states. It is important to note that the number of nest boxes occupied with one or several litters in a given month never exceeded 50% even at high population densities, suggesting that the observed increase in the propensity to raise litters communally was not simply a consequence of nesting site limitation.

**To breed communally or not**. To assess the effect of communal versus solitary breeding on the population dynamics, we constructed and analysed a set of matrix population models that were parameterised with the observed litter sizes, and survival and breeding probabilities as estimated using the MSCMR analyses, where we fixed the probability of raising a litter communally to 1 (all litters raised communally) or 0 (all litters raised solitarily). We, thus, simulated two hypothetical populations in which all females raised their litters either solitarily or communally, while keeping the overall survival and breeding probabilities the same. This allowed us to compare the population growth rate ($\lambda$, also a measure for fitness) between populations using only one of the two alternative breeding tactics. Both at low and high densities, a purely solitary breeding tactic outcompeted a communally breeding tactic (Fig. 3). The difference in $\lambda$ between a purely solitarily and a purely communally breeding population, however, was more pronounced, and different from 0, only at high (mean of the difference [95% CI]: 0.16 [0.10–0.22]) and not at low population densities (0.11 [−0.02–0.24]). The same can be seen in

the sensitivity analyses. Increasing the probability to raise litters communally in all states reduced the population growth rate $\lambda$ (negative sensitivities, Fig. 3), suggesting a negative effect of communal breeding on the population growth rate at higher densities.

Such a difference between high and low density could be explained by the density-dependent breeding and communal breeding probabilities. At high densities, females had overall a lower breeding probability and additionally more of those litters were communal (due to the higher probability to breed communally at high densities). The combined effect of a lower breeding probability and smaller average litter sizes because of more communal litters therefore caused the overall reduction in population growth rate. Nevertheless, communal breeding increased steadily, ranging from 40% in 2007 to 78% of all litters in 2014 (Fig. S1, SI files), therefore contributing to the slower population growth at higher densities.

At an individual level, communally breeding females had smaller litter sizes at weaning, and were not more likely to produce another litter the next month compared to solitarily breeding females. These results support a previous study showing a lower lifetime reproductive success for females raising a larger proportion of their litters communally[5], but contradict a previous lab study that found direct fitness benefits for communally breeding females[21].

Overall, our findings point towards a detrimental effect of communal breeding on the population growth rate. Similar

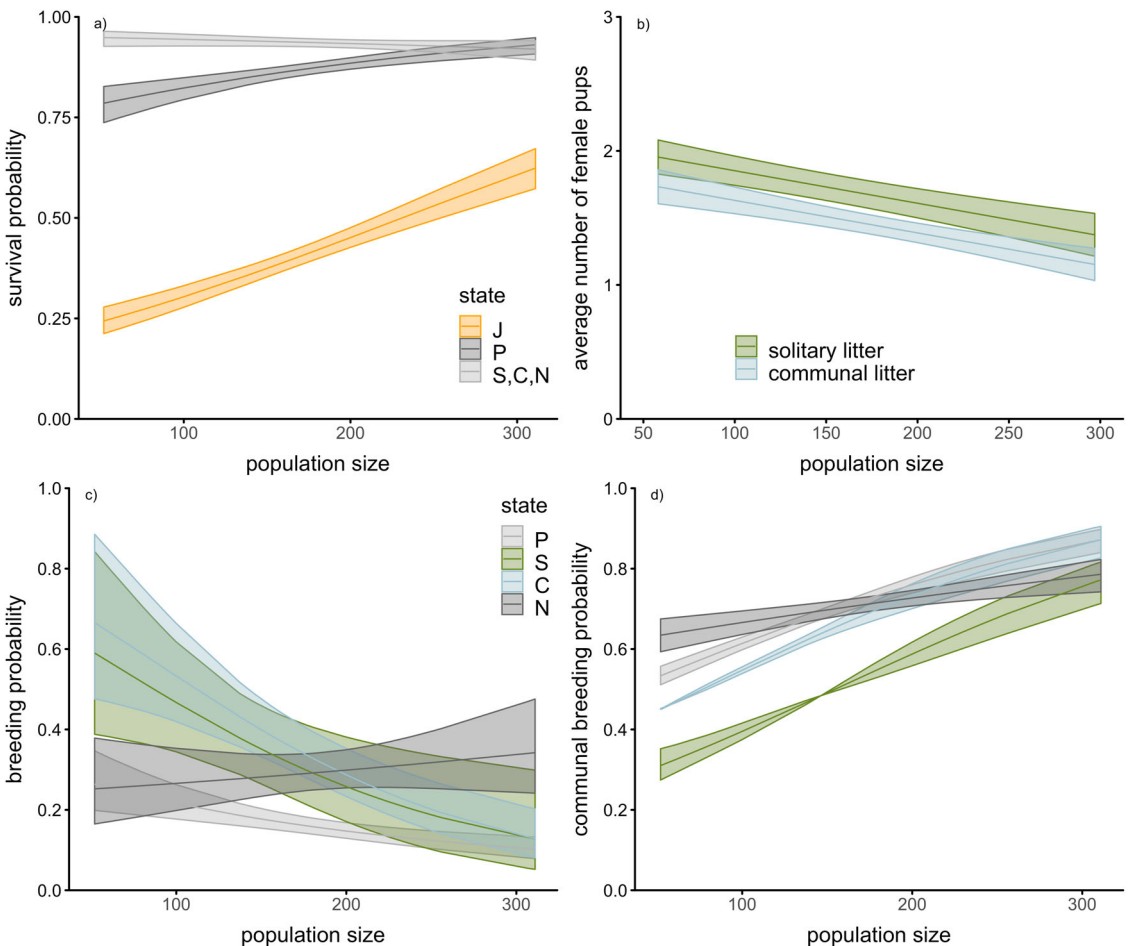

**Fig. 2 Density-dependent survival and breeding in female house mice during the breeding season (March to Semptember). a** Plotted are the state-dependent survival estimates and 95% CI from a multistate capture-mark-recapture model (for the years 2007 to 2014). States were defined as: J = juvenile, first month of age; P = prebreeder, second month or older, not yet breeding; S = solitary breeder, female raises a litter solitarily; C=communal breeder, female raises a litter communally; N = nonbreeder, female has no current litter but bred previously. **b** The average number of female pups weaned for litters raised solitarily or communally. **c** State-dependent probability to breed in the next month conditional on survival (estimates and 95% CI from a multi-state capture-mark-recapture model, abbreviations as before). **d** State-dependent probability to breed communally, conditional on breeding and survival (estimates and 95% CI from a multistate capture-mark-recapture model, abbreviations as before).

negative effects of communal breeding were also shown in other rodents, with the per capita reproductive success decreasing with each additional female in the group[8,47].

**Effect of female condition on breeding tactic**. The detrimental fitness consequences of communal breeding only apply if females that bred communally at a given time could have bred solitarily instead. This, however, may not always be the case. When reproductive competition is high (high population density, limited nesting sites) or when females are in poor physical condition, they may have to choose between breeding communally and not breeding at all. Body mass affects reproductive success in several species of mammals[48,49] and serves as a proxy for body condition. Females in poor condition should therefore be especially prone to communal breeding, and the propensity to breed communally should increase with increasing population density. Only highly competitive females, of high body mass, should be able to successfully raise litters solitarily at high densities.

Using generalised linear mixed models, we tested our expectations regarding the effect of body mass (as a proxy for condition or competitiveness) on a female's breeding probability the following month (conditional on survival), and the probability of raising a litter communally (conditional on breeding).

Body mass strongly influenced the breeding probability of prebreeders. With increasing body mass, primiparous females were more likely to have a litter the next month ($\beta = 1.60$ [1.03–2.17]; Fig. 4a). Furthermore, heavier females were less likely to raise a litter communally the next month ($\beta = -0.74$ [−1.40–−0.15]), and this was independent of their current breeding status (Table S5, SI files). With increasing population density, females were more likely to raise a litter communally ($\beta = 1.18$ [0.68–1.76]) and lighter females were more strongly affected by increasing population density compared to heavier females (interactive effect between population density and body mass, $\beta = -1.61$ [−2.78–−0.58]; Fig. 4b). These results are consistent with density- and condition-dependent breeding tactics[50,51]. When analysing the effect of density and a female's life-history state on her body mass, we also found a trend towards solitarily breeding females being heavier at higher densities, while density negatively impacted the body mass of communally breeding females (Fig. S4, SI files). This further supports our conclusion that only females in very good condition were able to raise litters solitarily at high population density.

**Concluding remarks**. Optimal reproductive strategies that maximize fitness in the face of constraints and associated trade-offs

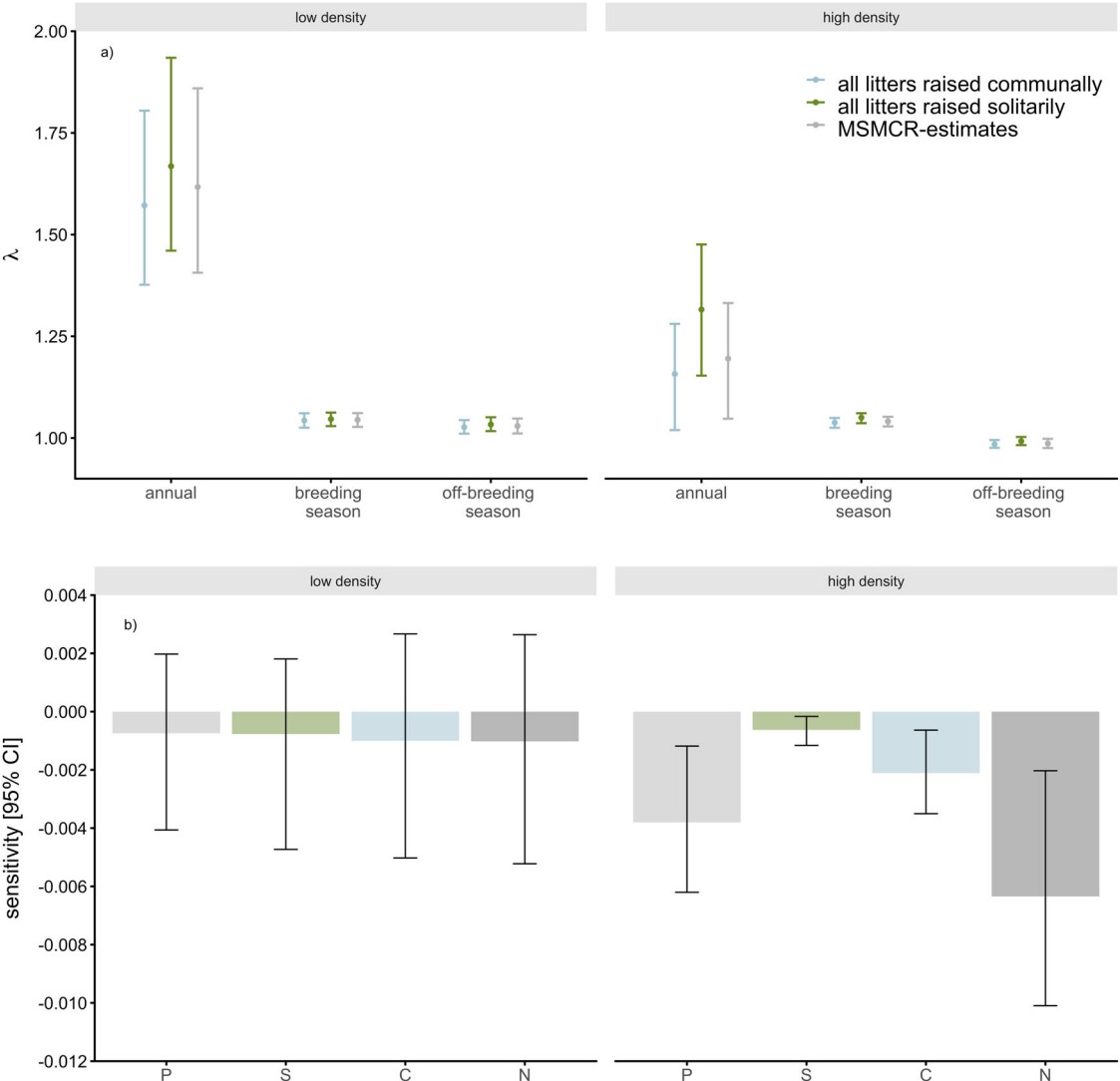

**Fig. 3 Matrix population models. a** Population growth rates ($\lambda$, [95% CI]) are plotted for season- and density-specific matrix population models that were parameterised with the estimates from the MSCMR-models in grey. In blue and green, $\lambda$ estimates from matrix models with the same survival and breeding values are shown, but the probability to breed communally is fixed to 1 (blue) or 0 (green), simulating populations with females using only one of the two tactics. **b** Sensitivities [95% CI] for the probability to breed solitarily during the breeding season at two different population densities. The Matrix population models were again parameterised with the estimates from the MSCMR-model (see Table S3, SI).

are thought to evolve from the interplay between an individual's intrinsic state and its extrinsic circumstances[50,52]. Our results revealed that breeding decisions of female house mice are condition- and density-dependent and that the alternative reproductive tactics of solitary and communal breeding have substantially different population-dynamic consequences. Females raise litters solitarily if they are in good physical condition (high body mass) or if there is little reproductive competition (low population density), but use a communal breeding tactic if they are not in good enough condition to successfully breed or defend their nest alone against competing intruders. Thus, alternative, condition-dependent breeding tactics allow female house mice at a subpar condition to breed when reproductive competition is high and they might otherwise not be able to breed. The persistence of communal breeding behavior in our study population, despite its costs, might represent a "best of a bad job" tactic[1,5]. The observed increase in the proportion of the communally-raised litters does therefore not indicate directional selection for communal breeding. Instead, it simply reflects the fact that at high densities reproductive competition, potentially

driven by increasing rates of infanticide, results in only the most competitive females in excellent physical condition being able to raise litters solitarily. Such condition-dependent alternative breeding tactics can be evolutionary stable even if they do not result in equal fitness[1,51]. Optional communal breeding has been described for other small mammals such as meadow voles (*Microtus pennsylvanicus*,[53]), degu (*Octodon degus*,[8]) and striped mice (*Rhabdomys pumilio*,[54]), without information on the behavior's fitness consequences.

Our study revealed that alternative breeding tactics can mediate the costs of reproductive competition at the population level and result in growing populations even under very crowded conditions and with many females in relatively poor condition. Individual behavioural plasticity dependent on ecological and social conditions resulted in population growth for both solitary and communal breeding. For house mice, this novel insight might explain the long-standing ecological puzzle of population outbreaks as has been frequently observed in Australian farms[55], during which mice reach very high densities. Our study further illustrates that laboratory studies analysing the costs and benefits

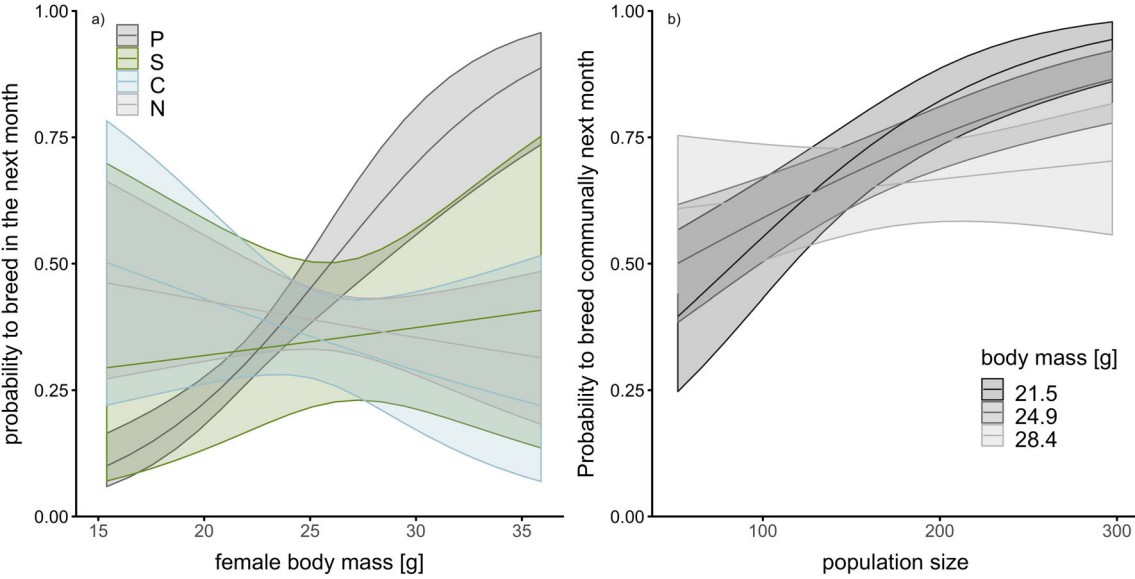

**Fig. 4 Body mass-dependent breeding probabilities. a** The effect of body mass on a female's probability to breed the next month (conditional on survival) and **b** the interactive effect of body mass and population size on a female's probability to breed communally the next month (conditional on breeding). For abbreviations of life-history states see legend of Fig. 1. Plotted are model estimates and 95% CI from generalised linear mixed models.

of alternative reproductive tactics might lead to erroneous conclusions[21]. Standardised conditions usually do not provide an environment with age- or life-history state-dependent survival and breeding, and thus do not necessarily allow for conclusions regarding the adaptiveness of a specific social behavior. Our results therefore highlight the importance of studying cooperation in the context of alternative reproductive behaviour in free-living populations. Analyzing the effect of reproductive decisions on population growth rates under natural conditions will contribute to our understanding of the evolution of alternative breeding tactics overall and likely reveal more situations where tactics represent a "best of a bad job" situation.

## Material and Methods
**Study population.** Data were collected from January 1, 2007, until December 31, 2014, as part of a long-term project on free-living house mice, near Zurich, in Switzerland. The population inhabited a former agriculture barn (72 m²) and was provided with food and water all year round. Forty artificial nest boxes served as breeding sites and straw and hay were provided as nest-building material. The set-up was modelled after the conditions of commensally living house mice in western Europe, which is why food and water were provided *ad libitum*. The building was closed against larger predators, while still allowing house mice, and other small animals (other rodents, shrews, small predators), to enter and leave freely. The mice reproduced all year round, but the majority of litters (92%) were born between March and September, which we classified as the breeding season and October to February therefore as off-breeding season. Nest boxes were checked for new litters at least every 13 days, with number of pups and age recorded. When pups were 13 days old (range: 11–14 days), they were sexed, weighed and a tissue sample (ear punch) was collected for later genetic analysis. From an age of 17 days onwards, house mice pups are weaned and they become fully independent at 23 days.

The entire house mouse population was censused every 6 to 8 weeks (in capture events). All captured mice were weighed, and individuals reaching a threshold body mass of ≥17.5g were subcutaneously injected with a transponder (RFID tag; Trovan ID-100A implantable micro-transponder: 0.1 g weight, 11.5 mm

length, 2.1 mm diameter) for individual recognition. In addition, their sex and reproductive status were determined, for females whether they appeared pregnant or lactating. We used the numbers from the capture events to estimate the minimum number of adult mice alive (total, as well as by sex and reproductive status) in a given month, which we used as a measure for population density. The population density is the same as the total population size, given that the area of the barn remained constant over the study period. All mice found dead were either identified based on their RFID tag, or through genetic analyses (see below). Antennas at the entrances of all nest boxes recorded the movement of individuals entering and leaving the nest boxes (recording of their RFID identity by the antennas). We used that information to record the period of time an individual was present in the barn (for details about the antenna system see König et al.[34], König and Lindholm[26]). Tissue samples were collected (ear punch collected from pups, newly tagged and dead individuals) allowing us to match pup and adult genotypes and to conduct parentage analyses. Markers at 25 polymorphic microsatellite loci were used to conduct the parentage analyses[56,57]. We used the program CERVUS 3.0 to assign a mother and a father to each pup[58]. All females and males recorded in the barn during the 30 days prior to the birth of a pup were included as potential parents, unless they were recorded as having died before the birthdate (in case of females) or the estimated conception date (in case of males). Same-aged pups that shared the same mother were assumed to be litter mates and this information was used to calculate litter size at sampling. Fecundity was estimated based on pups that survived to at least 11 days and therefore also includes early offspring survival. See König and Lindholm[34] and Ferrari et al.[5] for a more detailed description of the study site and methods used.

**Estimating survival and transition probabilities.** We used five age- and reproductive status-dependent states to describe the female portion of the population (n = 4003 females). Females that had not yet reproduced were either juveniles (in their first month of age), or prebreeders (in their second month or older), while mice that were currently breeding were classified as solitary (S) or communal (C) breeders, if they raised a solitary or a communal

litter, respectively, in a given month. Females that had previously reproduced, but had no litter in the current month were categorised as nonbreeders (N) (see Fig. 1 for a detailed life-cycle, showing all five states and the possible transitions between the life-history states). We used a multi-state capture-mark-recapture (MSCMR) model to estimate state- and season-specific recapture ($p_i$), survival ($\sigma_i$), and transition probabilities ($\psi_{j,i}$) from one state to another[35,37,59,60].

The model was fit using MARK[61], through the RMark interface[62] (R-version 3.4.2). We divided the entire study period (January 2007–December 2014) into monthly sessions (96 sessions) and documented for each female whether she was observed in the population during a given session (either by the antenna system or recorded during a population capture event) and determined the state (i) of the female. If a female had given birth at any point within a month, she was considered as breeder and was further classified as a solitary (S) or a communal (C) breeder. Gestation time in house mice is minimally 19 days (interbirth interval in the population is 66.7 ± 2.8 days (mean ± SE),[5]) and in a few instances (73 out of 1836 litters) a female had given birth twice within a single calendar month (i.e. within a single session). We excluded those females from the data, given that they made up only 4% of all observed litters. If a female was not observed in the population in a given month, even though the antenna system was working and/or a population capture event had taken place, we classified her as not encountered (0). Sessions during which no population capture event took place and the antenna system was not working were treated as missing values (.). Juveniles could only be encountered once, and their recapture probability ($p_J$) was therefore set to zero. Similarly, juveniles that survived their first month would automatically become prebreeders and the transition $\psi_{J,P}$ was therefore fixed to one.

We ran a set of biologically meaningful candidate models containing different covariates for $p$, $\sigma$ and $\psi$ to determine the model structure that best described the population. The Akaike information criterion, adjusted for small sample sizes (AICc), was used to select the most parsimonious model[63]. A complete model selection table with all models considered in this study can be found in the SI files (Table S1, S2). When including season (breeding versus off-breeding) as a covariate for $\psi$, we fixed the transitions from S or C to either S or C to zero in the off-breeding season, to avoid running into model convergence issues, due to insufficient data for those transitions.

**Fecundity**. A linear model was used to estimate the average number of female pups ($f_C$ and $f_S$) for the two breeding tactics, in the breeding and off-breeding seasons at two different population densities. The age of the pups at the time the litter size was taken (our approximation for weaning) was used as a covariate to account for potentially confounding effects of age at sampling (mean ± SE sampling age: 12.83 ± 0.028 days, n = 1836 litters) on litter size.

**Matrix population model**. Using the estimated survival and breeding probabilities, we constructed a stage-structured, seasonal matrix population model (after[38]) for the female segment of the population to estimate asymptotic seasonal and annual population growth rate (i.e., average fitness), and the sensitivity values. The annual matrix model was built from 12 monthly matrix models, seven for the breeding and five for the off-breeding season. We assumed a post-breeding census, and fertility and survival estimates were calculated accordingly. Females were characterised as being in one of five distinct states (see Fig. 1 for a detailed life-cycle and the description of the states). The same life cycle can be expressed in the form of a population projection matrix (see equation in Fig. 1). The probability to breed communally ($C_i$) for a given state was taken to be conditional on the probability to breed ($B_i$). The asymptotic population growth rate and sensitivities of $\lambda$ to vital rates were calculated using the R package popbio[64]. Confidence intervals for $\lambda$ and sensitivity measures were estimated via data bootstrapping using 200 iterations[38].

**Testing for the effect of female condition on her breeding probability**. We used body mass as a proxy for female condition and tested for its effect on female breeding tactics. Body mass and age are correlated in this population[5], and we cannot completely separate them; however, body mass influences lactation[29] and seems therefore a good indicator for overall body condition. Data on female body mass were not available for all females and during all sampling sessions. Body mass was recorded during population census events every 6 to 8 weeks, and body mass data for pregnant females were excluded to avoid confounding effects, which left us with 1033 body mass measurements (from 579 different females). The incomplete body mass measurements did not allow us to incorporate body mass in the MSCMR analysis, which does not allow missing values for individual covariates. We therefore used generalised linear mixed models (GLMMs) with binomial error structure to first, test for the effect of female body mass in a given month on her breeding probability the following month (transitions to either S or C) and second, among those females, to test for body mass effects on the probability to transition to communal breeding. Female life-history state and population density were included as covariates. The full model with singular and additive/interactive effects, and all lower-level models were computed, and we chose the most parsimonious model based on AICc[65]. Female identity was used as a random effect to correct for repeated measures on the same female. We used the lme4 package[66] to run GLMMs with a binomial error structure and a logit link function in R[67]. Models were tested for over-dispersion and parametric bootstrapping was used to compute 95% CI to assess the significance of fixed effects.

**Reporting summary**. Further information on research design is available in the Nature Research Reporting Summary linked to this article.

## Data availability
Data used for the MSCMR analyses (Supplementary Data 1, 2 and 4) and to assess the effect of body condition on breeding probabilities (see Supplementary Data 2) are included in the SI files. All other data are available in the manuscript itself (or from the corresponding authors on reasonable request).

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

## Acknowledgements

We thank all former and current members of the mouse group for the data collection. Furthermore, we would like to especially thank Jari Garbely for genetic lab work and genotype scoring and Gabriele Stichel, Sally Steinert and Bruce Boatman for animal care. This study in particular was funded by the Swiss National Science Foundation (grant P2ZHP3_178017 to MF) and the long term data collection on wild house mice was financially supported by the Claraz Stiftung, the Foundation for Research in Science and the Humanities at the University of Zurich, the Promotor Stiftung, the Julius-Klaus Stiftung and the Swiss National Science Foundation (grant 31003A_176114, 31003A_120444 and 310030M_138389).

## Author contributions

M.F., B.K., M.O., and A.O. devised the study; M.F. processed and analysed the data with help from M.O. and A.O. M.F., A.K.L., and B.K. contributed to data collection; B.K. initiated and maintains the long-term study; A.K.L. maintains the genetic data; M.F. wrote the first version, all authors discussed the results and contributed to the final manuscript.

## Competing interests

The authors declare no competing interests.

## Ethical notes

The data collection and all methods used for the presented study were approved by the Swiss Veterinary office (Canton Zurich, license numbers 215/2006 and 51/2010).
