## [Peer Review File · Communications Biology]

Reviewers' comments:

Reviewer #1 (Remarks to the Author):

This manuscript describes a fascinating study with remarkable sample sizes. The long-term data and large samples permit population modeling and analysis that few studies can match. That said, many of the most interesting conclusions were anticipated by the earlier paper in the *American Naturalist* on the same topic: alternative reproductive tactics (ARTs) of females. The current manuscript should make clear in the statement of purpose what this study adds, and if there are new conclusions. Many of what seemed to be new conclusions had already been made in the previous paper. The rationale for this manuscript should clearly describe the NEW goals, and the reader should not have to go to the previous literature to figure out what is new and what is perhaps stronger support for previous conclusions. Conclusions that are unique to THIS study should be clearly identified.

I am not convinced that the "population fitness" analysis adds anything to the study. It appears that the units of selection are the individual and the breeding group; the population is highly unlikely to be a unit of selection and the data are not appropriate for analysis of population selection. Each female might adopt the "optimal" tactic for their density and kinship (see below) environment. So how important is it to compare lambda for groups of solitary and communal breeders? 50% of females switch these tactics between litters, so the alternative ARTs are not alternative traits of females; females show both, likely dependent on their own condition, their social environment, and their demographic environment. So, what is lambda actually telling us about the ARTs? Their demographic associations perhaps, but not the fitness of an inherited trait. The "trait" is clearly a GxE interaction, the shape of which is not clear, but would make a very interesting further study.

It might be argued that a female would "prefer" in some sort of fitness sense to be in great condition and breed solitary. This is obviously not a good way to think. The fact that females show both ARTs suggests fitness benefits of both under perhaps different environmental conditions (different social and ecological environments). And to make "fitness" an important topic in a study like this, I think some indication of underlying genetic variation should be apparent, either in the results or citations. After all, if there is no heritability, then measures of fitness are only appropriate to show potential for natural selection. I honestly think that there should be a "higher bar" (some deeper thinking) for suggesting fitness advantages where the trait is highly plastic.

The breeding groups are likely kin-based (see Dobson et al. 2000 – *CJZ* 78, 1806-1812; I really think citing this study might be helpful, as it presages both kin groups and especially the conditions for influences of infanticide that are brought up in this study; several Koenig studies point in this direction too). Little information about kinship with respect to communal breeding females is given or even cited, a weakness for this study. Again, since 50% of females switch between solitary and communal breeding, there may be a demographic influence on the presence of close kin (the social environment), and thus for the option of communal breeding. I also wonder whether females that are in poor body condition and breeding communally are lower in mass because they have to spend so much social time. Finally, it wasn't clear to me when infanticide occurs. My own observations are that it can occur either early or very late in weaning (thus influencing maternal body mass dynamics). Was there any indication of a demographic influence of reproductive suppression?

The manuscript was well written and well organized. I made a few comments on the manuscript, and attach a copy. I hope my comments above and on the attached are helpful.

All the best,

F. Stephen Dobson
fsdobson@msn.com

Reviewer #3 (Remarks to the Author):

This paper addresses an interesting question using a formidable database. My only main concern is that I am not sure of how the analyses were done. It appears that analyses of survival and state transition were done on a monthly basis, presumably using calendar months, from the first to the last day of each month. If that is the case, how did the analysis deal with a female that gave birth, say, on June 20 and was nursing pups in the nest until July 10? Would she be considered breeder twice, for the same litter? Would she then inflate lambda? Parts of the ms refer to 'her next litter' – but the analysis seems to be by month, not by litter or breeding opportunity for each female. Was the transition to a new litter accounted for? If this aspect can be better explained and accounted for, the results could be accepted with greater confidence.

Specific comments:

L. 11: Best to avoid the word 'quality' - this actually refers to body mass, so why not call it body mass?

L. 24: missing 's' after apostrophe. Actually, this would work by just deleting the apostrophe.

Table 1: This legend needs to be better explained, the table is unclear without a presentation of the intervals over which survival and breeding state were measured. When is 'status' assessed? How can a solitary breeder be a communal breeder, and how can 'breeders' not have a breeding probability of 100%. Some females were both solitary and communal breeders, but presumably that is because there are multiple breeding attempts in a season? If those details are not explained, the table is confusing. Perhaps a reference to fig. 1 would help.

L. 122: Could this be partly due to an increase in average age of monitored females over time, while density was increasing?

L. 134-138: This is a sensible interpretation, but are there any data on infanticide in solitary and communal litters? This statement also seems to contradict the Conclusion about relevance to mouse plagues in Australia.

L. 151: What was the sensitivity of lambda to the probability of communal breeding at high density?

L. 158: Here we are told about 'next litter' but most of the presentation appears to be about 'next month'. This needs to be clarified - did the analysis distinguish between litters?

L. 182: Sorry, I do not understand the logic here - would this not be the case also for solitary litters?

L. 216-219: Very interesting. I wonder whether this result may be due not to an overall increase in mass by S females, but by fewer light females going for the S strategy at high density?

L. 232-235: A potential test of this idea would be to look at the success of S females of below-average body mass at low and high density.

L. 234-236: But earlier the paper states that infanticide risk did not differ by breeding strategy. So how can infanticide select for communal breeding? Are there any other physiological/energetic advantages?

L. 247-249: I do not agree with this speculation because the data presented here do not show a decrease in energetic cost of communal compared to solitary breeding. If small females that breed solitarily had lower fitness than small females that breed communally, this speculation would be supported.

L. 278: Any idea of the average proportion of mice that were captured, and if it varied with season? Readers will need some assessment of the reliability of this measure of density.

L. 315-318: So each calendar month was set as an interval. Does that mean that for some females the 'transition' from one breeding status to the same breeding status actually reflected the fact that she was nursing the same litter at the end of one interval and at the start of the following interval?

L. 337-340: Again - what happens if a female is recorded as breeding in 2 months, but for the same litter? Would that inflate the estimate of lambda? How likely is this to happen? If pups are nursed for 17-23 days, presumably many females lactate over 2 consecutive months?

Fig. 1: I am still not totally sure I understand what was done. Survival was estimated from one month to the next, and the same for state transitions? If so, how were months established (calendar months?), and what would happen if a female's breeding episode overlapped 2 months? I also do not understand the statement 'a post-breeding census was assumed' given that multiple litters could be produced in a breeding season - was a definite cut-off between breeding and non-breeding seasons imposed on the analysis? I also do not understand why the figure shows a fertility (F_n) for non-breeders.

Marco Festa-Bianchet

Response to Reviewer's Comments:

Reviewer #1 (Remarks to the Author):

This manuscript describes a fascinating study with remarkable sample sizes. The long-term data and large samples permit population modeling and analysis that few studies can match. That said, many of the most interesting conclusions were anticipated by the earlier paper in the *American Naturalist* on the same topic: alternative reproductive tactics (ARTs) of females. The current manuscript should make clear in the statement of purpose what this study adds, and if there are new conclusions. Many of what seemed to be new conclusions had already been made in the previous paper. The rationale for this manuscript should clearly describe the NEW goals, and the reader should not have to go to the previous literature to figure out what is new and what is perhaps stronger support for previous conclusions. Conclusions that are unique to THIS study should be clearly identified.

Thank you, Dr. Dobson, for all your helpful comments and suggestions. We tried our best to address all of your concerns.

As you correctly pointed out, Ferrari et al. (2019; American Naturalist) addressed questions related to ART using a subset of the data. However, the focus of that paper differed in many ways from the current study and also suffered from important limitations (which we have addressed in the present manuscript). In Ferrari et al. (2019), we aimed to analyse whether communal and solitary breeding can be considered alternative reproductive tactics (ART), whether they affect a female's lifetime reproductive success as well as the survival of solitary and communal litters. We therefore did focus on breeding females only. Furthermore, we did not analyse other important life-history traits, such as breeding probabilities, the timing of birth, the order in which females raised their litters (communal and then solitary or vice versa). Given that the previous study was based only on breeding females, females that were currently not breeding, or had never bred were ignored. At any point in time a substantial proportion of females are not breeding; by ignoring non-breeder females we missed out valuable information on traits that are under natural selection. Also, this omission of the nonbreeding segment of the population did not allow us to study the full life cycle.

By using state-dependent capture-mark-recapture (CMR) and matrix population modelling frameworks, we could investigate all those missing aspects of house mouse life history in the present study. This approach allowed us to estimate state-specific survival, and more importantly, transition probabilities among all breeding states (solitary and communal breeders, and nonbreeders). The state dependent survival and breeding probabilities, transition among alternative breeding states as well as the finding that population density and body mass (as a proxy for condition) influenced females' breeding tactics were therefore all new findings in the present study. Furthermore, this study quantified the overall population consequences of different breeding tactics, which is entirely new.

We emphasized the new aspects of the current study by modifying the abstract (deleting reference to a "best-of-a-bad-job" strategy) and more clearly highlighting our aims in the introduction (lines 85-87).

I am not convinced that the "population fitness" analysis adds anything to the study. It appears that the units of selection are the individual and the breeding group; the population is highly unlikely to be a unit of selection and the data are not appropriate for analysis of population selection. Each female might adopt the "optimal" tactic for their density and kinship (see below) environment. So how important is it to compare lambda for groups of solitary and communal breeders? 50% of females switch these tactics between litters, so the alternative ARTs are not alternative traits of females; females show both, likely dependent on their own condition, their social environment, and their demographic environment. So, what is lambda actually telling us about the ARTs? Their demographic associations perhaps, but not the fitness of an inherited trait. The "trait" is clearly a GxE interaction, the shape of which is not clear, but would make a very interesting further study.

We fully agree with Dr. Dobson that the population is not the unit of selection, and this is not what we were trying to say. Instead, we are interested in the population-dynamic consequences of the choice of a particular breeding tactic.

The CMR and mixed model analyses revealed that the choice between the two breeding tactics was condition-dependent (influenced by population density and body condition). Estimated asymptotic population growth rate for hypothetical purely solitary breeding and purely communally breeding populations, allowed us to quantify

population-level consequences of fixed breeding tactics. Lambda can be regarded as the average fitness for a given strategy (censu Caswell 2001) and can therefore be used to compare the relative fitness of alternative strategies.

We found that communal breeding reduces the overall population growth rate. However, communal breeding by females under adverse conditions (poor body condition and high population density) would allow females to breed that would otherwise not be able to breed at all due to poor condition and increased reproductive competition. We made sure to clarify this throughout the manuscript (for example, lines: 85-87, 195-200, 263-265).

It might be argued that a female would “prefer” in some sort of fitness sense to be in great condition and breed solitary. This is obviously not a good way to think. The fact that females show both ARTs suggests fitness benefits of both under perhaps different environmental conditions (different social and ecological environments). And to make “fitness” an important topic in a study like this, I think some indication of underlying genetic variation should be apparent, either in the results or citations. After all, if these is no heritability, then measures of fitness are only appropriate to show potential for natural selection. I honestly think that there should be a “higher bar” (some deeper thinking) for suggesting fitness advantages where the trait is highly plastic.

In order to understand how the two tactics fluctuate in the population under different environmental conditions, we had to estimate the GxE effects. This would, as you describe, require the estimation of heritability and the level of genetic response to selection. This is beyond the scope of our study. Yet, by only estimating the average fitness, we are approximating the selection pressure on the two tactics, which in itself gives an important insight into selection acting on ART regardless of its heritability and plasticity. And as noted above, we were mainly aiming to look at the population consequences of the choice of the ART.

The breeding groups are likely kin-based (see Dobson et al. 2000 – CJZ 78, 1806-1812; I really think citing this study might be helpful, as it presages both kin groups and especially the conditions for influences of infanticide that are brought up in this study; several Koenig studies point in this direction too). Little information about kinship with respect to communal breeding females is given or even cited, a weakness for this study. Again, since 50% of females switch between solitary and communal breeding, there may be a demographic influence on the presence of close kin (the social environment), and thus for the option of communal breeding. I also wonder whether females that are in poor body condition and breeding communally are lower in mass because they have to spend so much social time. Finally, it wasn't clear to me when infanticide occurs. My own observations are that it can occur either early or very late in weaning (thus influencing maternal body mass dynamics). Was there any indication of a demographic influence of reproductive suppression?

Kinship does certainly play a role; females in groups with higher overall relatedness were shown to be more likely to communally breed (Harrison et al. 2018). We have included a paragraph where we discuss the potential effect of kinship in communally breeding house mice and included the mentioned study. We have also cited Dobson et al. (2000 – thanks for suggesting it (lines 52-58).

Communally breeding females overall experienced reduced litter sizes but breeding with a relative would be less costly than breeding with unrelated females – assuming both females could not have bred solitarily instead.

Infanticide in this study was measured indirectly by looking at litter size at around d11-d14, and thus covering early infanticide. Furthermore, juvenile survival was estimated as the survival probability of pups to the next month. Litter sizes decreased with increasing density, pointing towards more infanticide at higher densities. Juvenile survival on the other hand increased, as we discuss in the discussion section of the text. One potential problem is that we cannot distinguish between dispersal and death (unless corpses were found), which especially for young house mice could lead to biased survival probabilities.

At high densities breeding probabilities decreased across all states; reproductive suppression might therefore be at play. But again, we cannot disentangle the different effects. The observed reduction in breeding probability could be due to reproductive suppression of some females by others, due to a higher rate of complete early litter losses or due to aborted pregnancies.

The manuscript was well written and well organized. I made a few comments on the manuscript, and attach a copy. I hope my comments above and on the attached are helpful.

Thank you!

All the best,

F. Stephen Dobson
fsdobson@msn.com

Reviewer #3

This paper addresses an interesting question using a formidable database. My only main concern is that I am not sure of how the analyses were done. It appears that analyses of survival and state transition were done on a monthly basis, presumably using calendar months, from the first to the last day of each month. If that is the case, how did the analysis deal with a female that gave birth, say, on June 20 and was nursing pups in the nest until July 10? Would she be considered breeder twice, for the same litter? Would she then inflate lambda?

Thank you, Dr. Festa-Bianchet, for very helpful comments and suggestions. We have tried our best to address all of your concerns.

As you correctly point out, all of our analyses were conducted on a monthly time scale (the whole study period was divided into 96 distinct monthly sessions). House mice are continuous breeders and can give birth any day of the month. Nevertheless, we made sure that a female would not be considered a breeder in two consecutive months based on the same litters:

- If a female had given birth at any point within a calendar month, she was considered to be a breeder and was further classified as a solitary (S) or a communal (C) breeder depending on whether the litter was reared alone or communally.*
- Gestation time in house mice is minimally 19 days (average inter litter interval in the study population is 66.7 ± 2.8 days (mean \pm SE), Ferrari et al. 2019), and in most instances, females produced only one litter per month. In a very few instances (73 cases out of 1836 litters) females gave birth twice within a single calendar month. We excluded those females from our analyses to avoid the complications you pointed out. Given that they made up only 4% of all observed litters, exclusion of these reproductive events should not affect our results substantially.*
- We assigned breeding status of a female based entirely on giving birth to a litter (and not based on other criteria such as lactation). Thus, a female that gave birth on June 20 and was nursing pups in the nest until July 10 would be considered as a breeder for June. However, she would be considered non-breeding in July, unless she gave birth to another litter by the end of July. Therefore, a female could not have been recorded as breeding in 2 consecutive months for the same litter.*
- We now clarified this in the method section (lines 331-336)*

Parts of the ms refer to 'her next litter' – but the analysis seems to be by month, not by litter or breeding opportunity for each female. Was the transition to a new litter accounted for? If this aspect can be better explained and accounted for, the results could be accepted with greater confidence.

Because each month was treated as a new breeding opportunity, we could estimate females' likelihood to breed (solitarily or communally) in the following month, conditional on survival. We have made sure that this is clearly stated throughout in the revised version of the manuscript and avoid the use of "her next litter" now.

Specific comments:

L. 11: Best to avoid the word 'quality' - this actually refers to body mass, so why not call it body mass?

We changed this accordingly.

L. 24: missing 's' after apostrophe. Actually, this would work by just deleting the apostrophe.

We corrected it.

Table 1: This legend needs to be better explained, the table is unclear without a presentation of the intervals over which survival and breeding state were measured. When is 'status' assessed? How can a solitary breeder be a communal breeder, and how can 'breeders' not have a breeding probability of 100%. Some females were both solitary and communal breeders, but presumably that is because there are multiple breeding attempts in a season? If those details are not explained, the table is confusing. Perhaps a reference to fig. 1 would help.

We have revised Table 1 legend for clarity, as suggested. To clarify,

- Reproductive status of each female is assessed each month;*

- *State-specific breeding probability here refers to the probability that a female in state i this month breeds next month, conditional on survival. Because not all females that breed this month will breed next month (if they survive), breeding probability will be less than 100%;*
- *State-specific communal breeding probability here refers to the probability that a female in state i this month breeds communally next month, given that she survives until next months and breeds.*
- *Some females in state i this month can breed solitarily next month if they survive. However, a female can either be a communal or a solitary breeder at any given month (but not both).*

L. 122: Could this be partly due to an increase in average age of monitored females over time, while density was increasing?

This is an interesting question, but we found no evidence for an increase in average female age. There was no significant correlation between the monthly average age and time (Spearman correlation test: $\rho=0.15$, $p=0.13$).

L. 134-138: This is a sensible interpretation, but are there any data on infanticide in solitary and communal litters? This statement also seems to contradict the Conclusion about relevance to mouse plagues in Australia.

Most of the data about infanticide stem from laboratory studies. As we have eluded to in the Introduction, there are contradictory findings when it comes to the effect of communal breeding on infanticide avoidance. There is evidence that the female that gives birth second in a communal nest kills some offspring already in the nest, before giving birth herself (König 1994, Palanza et al. 2005, Schmidt et al. 2015, Ferrari et al. 2015; these references can be found in the manuscript). But there is also some evidence that there are no survival differences between communal and solitary litters after they were formed (Ferrari et al. 2019) or even a slight survival advantage for communally raised litters (König 1994, Manning et al. 1995, Auclair et al. 2014). In relation to the plagues in Australia, even though communal breeding does result in smaller litter sizes than solitary breeding, it still might allow mice to breed under conditions they normally would not be able to breed at all and therefore to further increase the population size.

L. 151: What was the sensitivity of lambda to the probability of communal breeding at high density?

Sensitivities of lambda to the probability of communal breeding at high density were [95% CI]:

- *Prebreeders: -0.004 [-0.006– -0.001]*
- *Solitary breeders: -0.001 [-0.001– -0.000]*
- *Communal breeders: -0.002 [-0.003– -0.001, and*
- *Nonbreeders: -0.006 [-0.010– -0.002]*

These values are illustrated in Fig. 3, we have now also included this information in the text (lines 156-158 and 161-165). Furthermore, see Table S4 for sensitivities of lambda to all parameter at low and high densities.

L. 158: Here we are told about 'next litter' but most of the presentation appears to be about 'next month'. This needs to be clarified - did the analysis distinguish between litters?

Our apologies for the confusion. By “next litter”, we meant the litter next month (assuming that they survive until next month and produce litters). We have rephrased the relevant sentences to eliminate “next litter”.

L. 182: Sorry, I do not understand the logic here - would this not be the case also for solitary litters?

Communal litters have smaller litter sizes at weaning. At high densities, females have overall a lower breeding probability and additionally more of those litters will be communal (due to the higher probability to breed communally at high densities). The combined effect of a lower breeding probability and smaller litter sizes because of more communal litter could therefore cause the overall reduction in population growth rate. We slightly rephrased it in the manuscript to hopefully clarify what we meant.

L. 216-219: Very interesting. I wonder whether this result may be due not to an overall increase in mass by S females, but by fewer light females going for the S strategy at high density?

Yes, this is what we suspect is happening and we discuss this in the text (lines 215-221, 244-248). Lighter females might not be able to rear litters solitarily at high densities, which is why we see an overall reduction in the probability of females to rear litters solitarily and why we find the correlation between female body mass and breeding tactic.

L. 232-235: A potential test of this idea would be to look at the success of S females of below-average body mass at low and high density.

This is an excellent idea, but unfortunately, difficult to test in free-living populations. We observe this correlation between body mass and what tactic females use and it is not possible to disentangle the two from each other. While we might have been able to identify some light females that bred solitarily and heavy females breeding communally, sample sizes would have been small and therefore more easily influenced by additional factors, such as the availability of other breeding females in the group. A controlled experiment in a semi-natural setting (with differences in density) might be required to test for such an effect.

L. 234-236: But earlier the paper states that infanticide risk did not differ by breeding strategy. So how can infanticide select for communal breeding? Are there any other physiological/energetic advantages?

As noted previously, the literature provides contradictory evidence regarding the effect of communal breeding on infanticide avoidance. In this particular instance, what we mean is that we see a negative effect of population density and a positive effect of female body mass on a female's probability to breed solitarily. In other words, communal breeding is more commonly observed under high reproductive competition. We can only observe litters that survive until at least day 11; it could be that under high competition and/or if females are of low body mass, they try to breed solitarily, but are unsuccessful because their offspring is killed before we can find them for the first time. Alternatively, females might not be able to prevent other females from adding their litters to the nest, and thus forcing them into communal breeding. Females of low condition and/or at high densities might therefore only be able to breed communally, or not be able to breed at all.

L. 247-249: I do not agree with this speculation because the data presented here do not show a decrease in energetic cost of communal compared to solitary breeding. If small females that breed solitarily had lower fitness than small females that breed communally, this speculation would be supported.

With our data, we cannot show that small females have lower fitness when they breed solitarily versus communally. However, we can observe that smaller females are less likely to breed solitarily, especially at high density, indicating that they are not able to breed solitarily under those conditions. Communal breeding would then be more beneficial than not breeding at all, despite the costs associated with it (i.e. smaller litter size at weaning compared to solitary litters).

L. 278: Any idea of the average proportion of mice that were captured, and if it varied with season? Readers will need some assessment of the reliability of this measure of density.

The most parsimonious multistate capture-mark-recapture model (Table S2) revealed that the state-specific capture probability (p) differed between season and was affected by population size, and generally ranged from ~0.3 for prebreeders at high densities to ~0.9 for adults (including nonbreeders, and solitary and communal breeders) at low density. Capture probabilities were higher for breeding females, slightly higher during off-breeding season, and were affected negatively by population density. We have now reported these results in the supplementary materials (see Fig. S2) and referred to it in the text (lines 95-96).

L. 315-318: So each calendar month was set as an interval. Does that mean that for some females the 'transition' from one breeding status to the same breeding status actually reflected the fact that she was nursing the same litter at the end of one interval and at the start of the following interval?

Please see our response to the next comment.

L. 337-340: Again - what happens if a female is recorded as breeding in 2 months, but for the same litter? Would that inflate the estimate of lambda? How likely is this to happen? If pups are nursed for 17-23 days, presumably many females lactate over 2 consecutive months?

As you correctly point out, all of our analyses were conducted on a monthly time scale. However, assignment of the breeding status of a female was based entirely on the birth of a litter; a female was considered to have bred in a month during which she gave birth to a litter. It was therefore impossible for a female to have been considered breeding for 2 months due to only one litter (see our earlier comment for a more detailed answer). This has now been clarified in the Methods section (lines 331-336).

Fig. 1: I am still not totally sure I understand what was done. Survival was estimated from one month to the next, and the same for state transitions? If so, how were months established (calendar months?), and what would happen if a female's breeding episode overlapped 2 months? I also do not understand the statement 'a post-breeding census was assumed' given that multiple litters could be produced in a breeding season - was a definite cut-off between breeding and non-breeding seasons imposed on the analysis? I also do not understand why the figure shows a fertility (F_n) for non-breeders.

We have addressed some of these issues above, and briefly summarize here:

- We used calendar months in all our analyses of survival and reproductive rates, as well as for the analysis of matrix population models. However, assignment of the breeding status of a female was based entirely on birth of a litter (and not based on other criteria such as lactation) – she would only be considered breeding in the month she gave birth to a litter.*
- We excluded a few females that gave birth twice within a single calendar month (see our earlier comment).*
- We estimated state-specific monthly survival rate as the probability that a female that is alive and in state i this month will survive to the next month.*
- We estimated state-specific breeding probability as the probability that a female in state i this month transitions to state j next month, conditional on survival.*
- Using the aforementioned survival and breeding/transition probabilities and litter size data, we calculated matrix model parameters assuming post-breeding census methods (Caswell 2001). Here, “post-breeding census” simply describes a situation where demographic data are collected just after the breeding event, which was the case in our study.*
- As you correctly note, females that are currently in non-breeding state can have non-zero fertility rate. This is because some females that are in non-breeding state can survive the month and can potentially breed at or before the end of the month.*

We have now clarified this in the Methods section and in figure legends (Fig. 1, lines 231-236).

REVIEWERS' COMMENTS:

Reviewer #1 (Remarks to the Author):

I reviewed this manuscript previously, and I'm very pleased with the improvements that were made with minimal changes. Although I disagree with the "this is fitter than that" approach of the authors, the text is now somewhat more cautious about the presentation of these ideas. The main point of the article is that knowing about the demography of alternative behaviors with respect to solitary versus social breeding is interesting and perhaps useful. The manuscript succeeds in this regard.

Perhaps the most interesting result of the study is that with alternative social reproductive modes, female mice can maintain positive population growth when the population is quite crowded and many females are in relatively poor condition (compared to when density is low). The alternative reproductive modes result from a behavioral reaction norm that all females likely exhibit (genetic variation among reaction norms would be a GxE interaction, but this is currently unknown). But it is clear that individual females can varied their reproductive mode according to social and ecological conditions, and both solitary and communal breeding resulted in positive population growth. This insight was not enough emphasized, in my opinion.

Nonetheless, this is a strong study of an extensive dataset. The authors have clearly worked hard and met my comments more than half way.

Best of luck, Steve Dobson (fsdobson@msn.com)

Reviewer #2 (Remarks to the Author):

Thank you for the clarifications, this is a very good paper. I just have 2 minor comments:

Table 1: Table legend should explain what J P S C and N mean, or refer to Fig. 1

Lins 278-279: Does this imply that small predators (weasels? shrews?) could enter the building?

Marco Festa-Bianchet

Response to Reviewer's Comments:

Reviewer #1 (Remarks to the Author):

I reviewed this manuscript previously, and I'm very pleased with the improvements that were made with minimal changes. Although I disagree with the "this is fitter than that" approach of the authors, the text is now somewhat more cautious about the presentation of these ideas. The main point of the article is that knowing about the demography of alternative behaviors with respect to solitary versus social breeding is interesting and perhaps useful. The manuscript succeeds in this regard.

Perhaps the most interesting result of the study is that with alternative social reproductive modes, female mice can maintain positive population growth when the population is quite crowded and many females are in relatively poor condition (compared to when density is low). The alternative reproductive modes result from a behavioral reaction norm that all females likely exhibit (genetic variation among reaction norms would be a GxE interaction, but this is currently unknown). But it is clear that individual females can varied their reproductive mode according to social and ecological conditions, and both solitary and communal breeding resulted in positive population growth. This insight was not enough emphasized, in my opinion.

Nonetheless, this is a strong study of an extensive dataset. The authors have clearly worked hard and met my comments more than half way.

Best of luck, Steve Dobson (fsdobson@msn.com)

Thank you, Dr. Dobson, for these additional helpful comments and suggestions. We tried our best to address all of your concerns and now more strongly emphasise your last point in the discussion section of the manuscript [lines 254-260].

'Our study revealed that alternative breeding tactics can mediate the costs of reproductive competition at the population level and result in growing populations even under very crowded conditions and with many females in relatively poor condition. Individual behavioural plasticity dependent on ecological and social conditions resulted in population growth for both solitary and communal breeding. For house mice, this novel insight might explain the long-standing ecological puzzle of population outbreaks as has been frequently observed in Australian farms [55], during which mice reach very high densities.'

Reviewer #2 (Remarks to the Author):

Thank you for the clarifications, this is a very good paper. I just have 2 minor comments:

Thank you, Dr. Festa-Bianchet, for your additional helpful comments and suggestions.

Table 1: Table legend should explain what J P S C and N mean, or refer to Fig. 1

We refer now to Figure 1 in the legend of table 1 to avoid any confusion about the used abbreviations.

Lines 278-279: Does this imply that small predators (weasels? shrews?) could enter the building?

Yes, small animals such as other rodents, but also shrews and weasels should be able to enter the barn (though to our knowledge no weasel was ever observed inside, however, there were sightings of shrews and wood mice. We added a clarification to the manuscript [line 278].